# Rapid Molecular Diagnosis of Genetically Inherited Neuromuscular Disorders Using Next-Generation Sequencing Technologies

**DOI:** 10.3390/jcm11102750

**Published:** 2022-05-12

**Authors:** Sofia Barbosa-Gouveia, Maria Eugenia Vázquez-Mosquera, Emiliano González-Vioque, Álvaro Hermida-Ameijeiras, Paula Sánchez-Pintos, Maria José de Castro, Soraya Ramiro León, Belén Gil-Fournier, Cristina Domínguez-González, Ana Camacho Salas, Luis Negrão, Isabel Fineza, Francisco Laranjeira, Maria Luz Couce

**Affiliations:** 1Unit of Diagnosis and Treatment of Congenital Metabolic Diseases, Department of Paediatrics, Santiago de Compostela University Clinical Hospital, 15704 Santiago de Compostela, Spain; maria.eugenia.vazquez.mosquera@sergas.es (M.E.V.-M.); alvaro.hermida@usc.es (Á.H.-A.); paula.sanchez.pintos@sergas.es (P.S.-P.); mj.decastrol@gmail.com (M.J.d.C.); 2Centro de Investigación Biomédica en Red de Enfermedades Raras (CIBERER), IDIS-Health Research Institute of Santiago de Compostela, Santiago de Compostela University Clinical Hospital, European Reference Network for Hereditary Metabolic Disorders (MetabERN), 15704 Santiago de Compostela, Spain; 3Department of Clinical Biochemistry, Puerta de Hierro-Majadahonda University Hospital, 28222 Majadahonda, Spain; egvioque@gmail.com; 4Genetics Department, Hospital Universitario de Getafe, 28905 Madrid, Spain; soraya.ramiroleon@salud.madrid.org (S.R.L.); belen.gilfournier@salud.madrid.org (B.G.-F.); 5Neuromuscular Unit, Imas12 Research Institute, Hospital Universitario 12 de Octubre, 28041 Madrid, Spain; cdgonzalez@salud.madrid.org; 6Center for Biomedical Network Research On Rare Diseases (CIBERER), Instituto de Salud Carlos III, 28029 Madrid, Spain; 7Pediatric Neurology Unit, Hospital Universitario 12 de Octubre, Complutense University of Madrid, 28041 Madrid, Spain; acamacho@salud.madrid.org; 8Neuromuscular Diseases Unit, Neurology Service, Centro Hospitalar e Universitário de Coimbra, 3000-075 Coimbra, Portugal; luisnegraoster@gmail.com; 9Pediatric Neurology Department, Child Developmental Center, Hospital Pediátrico, Centro Hospitalar e Universitário de Coimbra Coimbra Portugal, 3000-075 Coimbra, Portugal; ifineza@gmail.com; 10Biochemical Genetics Unit, Centro de Genética Médica Doutor Jacinto Magalhães, 4050-466 Porto, Portugal; francisco.laranjeira@chporto.min-saude.pt

**Keywords:** myopathy, neuromuscular disorders, next-generation sequencing, Phenomizer

## Abstract

Neuromuscular diseases are genetically highly heterogeneous, and differential diagnosis can be challenging. Over a 3-year period, we prospectively analyzed 268 pediatric and adult patients with a suspected diagnosis of inherited neuromuscular disorder (INMD) using comprehensive gene-panel analysis and next-generation sequencing. The rate of diagnosis increased exponentially with the addition of genes to successive versions of the INMD panel, from 31% for the first iteration (278 genes) to 40% for the last (324 genes). The global mean diagnostic rate was 36% (97/268 patients), with a diagnostic turnaround time of 4–6 weeks. Most diagnoses corresponded to muscular dystrophies/myopathies (68.37%) and peripheral nerve diseases (22.45%). The most common causative genes, *TTN*, *RYR1*, and *ANO5*, accounted for almost 30% of the diagnosed cases. Finally, we evaluated the utility of the differential diagnosis tool Phenomizer, which established a correlation between the phenotype and molecular findings in 21% of the diagnosed patients. In summary, comprehensive gene-panel analysis of all genes implicated in neuromuscular diseases facilitates a rapid diagnosis and provides a high diagnostic yield.

## 1. Introduction

Inherited neuromuscular disorders (INMDs) are a group of diseases with a variable age of onset, ranging from the neonatal period through to childhood to adulthood [1,2]. INMDs are both clinically and genetically heterogeneous and include a wide range of diseases that affect the muscular and peripheral nervous systems. INMDs include muscular dystrophies/myopathies, peripheral nerve diseases (e.g., Charcot–Marie–Tooth disease), motor neuron diseases (hereditary spastic paraplegias, spinal muscular atrophy, spinal-bulbar muscular atrophy), ion channel diseases (myotonia congenita, paramyotonia), and neuromuscular junction diseases (congenital myasthenic syndromes, myasthenia gravis) [2]. Common clinical features include progressive or episodic muscle weakness, muscle cramps and contractures, muscle pain and spasticity, cranial nerve palsies, and myotonia [3,4,5,6]. Circulating biomarkers, such as creatine kinase, antibodies, neurofilaments, and microRNAs, can facilitate diagnosis and prognosis in INMD patients, and help predict therapeutic response [7]. 

Establishing a definitive INMD diagnosis remains challenging. Many approaches can be used, each requiring careful clinical evaluation of the patient: (i) social or therapeutic drug use since these can induce myopathies or myasthenia [8,9]; (ii) musculoskeletal and neurologic examination; (iii) electromyography and nerve conduction studies [10]; (iv) measurement of creatine kinase levels, which is an indicator of muscle fiber breakdown, or of the presence of autoantibodies (as occurs in myasthenia gravis), which leads to abnormal neuromuscular transmission [11,12]; and (v) genetic testing. Recent advances in precision medicine and next-generation sequencing (NGS) technologies have helped to uncover the molecular basis of INMDs. In the last decade, 600 genes have been identified and associated with these disorders [13,14]. When there is a strong clinical suspicion of INMD but the patient presents with highly heterogeneous clinical signs or there are insufficient findings to guide a specific diagnosis, genetic testing options include targeted gene sequencing using panels specific for a group of molecularly well-characterized disorders [15,16], clinical exome sequencing (CES) [17], and whole-exome sequencing (WES) [18,19]. In addition to these approaches, neurophysiological, histological, and laboratory analyses remain important tools for confirming genetic findings [20].

In the present study, we evaluated the utility of a targeted gene panel to characterize the mutational architecture of INMD-associated genes, and quantified the diagnostic yield of this approach. We identify and discuss the main reasons for the persistent dearth of molecular data on several INMDs, despite the availability of NGS technologies. Moreover, we emphasize the importance of establishing and following an appropriate diagnostic strategy in order to achieve an early genetic diagnosis, thereby avoiding further costly procedures and enabling the implementation of treatment in early disease stages.

## 2. Results

Our study population consisted of 268 pediatric and adult patients with a suspected INMD, with the age of onset ranging from prenatal to adulthood. Females accounted for 111 patients (41.41%; mean age, 32.79 ± 21.89 years) and males 157 (58.58%; mean age, 40.04 ± 21.91 years). The median coverage achieved in the samples studied was 324.05X, with 98.17% target average coverage at 20X.

### 2.1. Diagnostic Rates

A total of 88 patients (33%) were diagnosed with an INMD, 44 (16%) had an inconclusive diagnosis, and no evidence of a molecular defect was detected in 136 (51%). To achieve a positive diagnosis, family studies were carried out to confirm disease segregation, except for those cases where the physician had confirmed the molecular diagnostic result (P41, P46, and P73). Of patients with an inconclusive diagnosis, a pathogenic/likely pathogenic variant closely related to the patient’s clinical phenotype was detected in a recessive gene in 23% (8/35) of cases but no second variant was detected; 2 possible cases of dual diagnosis were suspected in 6% (2/35) of cases; and in the remaining 71% of cases (25/35), variants of uncertain significance (VUS) were identified and family studies could not be completed, making it impossible to establish disease segregation. Detailed clinical information and corresponding genetic findings are presented in the Appendix A).

During the 3-year study period, 3 versions of the multi-gene panel were designed. The progressive addition of genes to the panel design resulted in an exponential increase in the rate of diagnosis. The first version, which included 278 INMD-related genes, yielded a rate of diagnosis of 31% while the third, which included 324 genes, yielded a rate of diagnosis of 40% (Figure 1). The mean diagnostic rate over the entire study period was 36%.

### 2.2. Genetic Findings

In our cohort, de novo variants were identified in 28 of the 97 diagnosed patients. The most common causative genes identified were ANO5 and TTN (9/97 patients), RYR1 (8/97 patients), DMD and SH3TC2 (7/97 patients), and COL6A3 and CAPN3 (5/97 patients) (Figure 2A). Most of the diagnosed cases corresponded to muscular dystrophies/myopathies (68.37%) and peripheral nerve diseases (22.45%) (Figure 2B). The diagnostic rate achieved in each INMD group was as follows: 46.53% in muscular dystrophies/myopathies, 30.99% in peripheral nerve diseases, 18.91% in neuromuscular junction diseases/ ion channel diseases, and 12.5% in motor neuron diseases.

Copy number variants (CNVs) were identified in six patients (Figure 3). A homozygous deletion of exon 4 in *SIGMAR1* was identified in P87, and a corresponding heterozygous compound deletion in P88, which in both cases affected the hydrophobic region that includes the cholesterol-binding domain (Figure 3A). In *DMD*, a hemizygous deletion was identified in P30, spanning exons 21–33, and a hemizygous duplication in P29, spanning exons 24–29, affecting the rod domain of the protein in both cases (Figure 3B). Finally, in *PMP22*, a deletion spanning exons 4 and 5 was detected in P73, and a duplication of the entire gene in P72 (Figure 3C).

In cases in which a positive molecular diagnosis was achieved, the diagnostic turnaround time was four to six weeks. For differential diagnosis, we used the Phenomizer algorithm to analyze data from the 97 patients for whom more complete medical histories were available to identify candidate disease-causing genes based on patients’ clinical features. In 21 of those cases (22%), Phenomizer produced a statistically significant *p*-value, based on which we established a correlation between the clinical features and molecular findings (Appendix A).

Among patients with an inconclusive diagnosis, six were found to carry a heterozygous pathogenic/likely pathogenic variant in a recessive gene closely related to the patient’s clinical phenotype, although no second variant was detected (Appendix A). Using Phenomizer, we obtained a statistically significant *p*-value in three of these cases: P91, P93, and P95. A dual diagnosis was suspected in 6% (2/35) of cases. P100 was a 5-year-old girl with homozygous variants in *TRAPPC11*, associated with limb-girdle muscular dystrophy, and *PYGM*, associated with McArdle disease. P101 was a 3-year-old boy with a hemizygous indel in *DMD*, associated with Duchenne muscular dystrophy, and a heterozygous missense variant in *MYH7*, associated with distal myopathy (Appendix A).

A gene expression network assessment was carried out using Reactome pathway enrichment analysis [21] in order to understand how the different levels of biological systems are associated in INMD (Figure 4). Most of the genes analyzed in this study are clearly related to muscle contraction, extracellular matrix (ECM), and the development of the nervous system (Table 1).

## 3. Discussion

Despite the increase in our knowledge of the molecular basis of INMDs, a high rate of definitive diagnosis remains difficult to achieve. In our study, targeted gene-panel sequencing yielded a molecular diagnosis in 88/268 (33%) patients with a suspected INMD. Recent studies reported diagnostic rates of 15–49.3% for targeted gene-panel sequencing [16,22,23,24,25], 19–62.9% for CES, [17,26,27], and 12.9–39% for WES [17,18,19]. Despite the existence of ACMG guidelines to better classify pathogenic variants, there are still conflicting interpretations among different genetic laboratories, which in turn affects the diagnostic rate. In a recent study, Winder et al. aimed to demonstrate the clinical utility of a targeted gene panel including 266 genes by analyzing 25,356 unrelated patients with a suspicion of INMD. A definitive diagnosis was achieved in 5055 (20%) of the patients, and the CNVs detection accounted for 39% of the variants identified [28]. Savarese et al. used an NGS platform named MotorPlex, including 93 genes associated with nonsyndromic myopathies, which usually cannot be clinically diagnosed. In total, 504 patients and 84 family members from the Italian Network of Congenital Myopathies and the Italian Network of Limb-Girdle Muscular Dystrophy were studied and as a result, 218 (43.3%) cases obtained a positive diagnosis, and 160 patients had an undetermined diagnosis since interesting candidate variants were identified but unproven [29].

A key genetic feature of INMDs is a high incidence of de novo variants [4], which we identified in 26 of the 97 diagnosed patients. This high rate might be associated with an early onset of the disorder (18 patients with a de novo variant were less than 20 years old) and the possibility to perform family studies. However, regardless the genetic testing strategy, the diagnostic rates reported in the literature remain more or less the same. There are several potential explanations for this observation: (i) a lack of complete family studies can lead to inaccurate interpretation of genetic testing results; (ii) identification of VUS does not guarantee a definitive diagnosis since functional studies are needed to validate any association with the disorder; (iii) although advances in NGS technologies have identified new disease-associated genes, molecular analysis and interpretation is still focused on variations in gene-coding regions; and (iv) clinical heterogeneity and overlapping phenotypes are characteristics of INMDs, in which both the clinical presentation and laboratory findings are often nonspecific. Furthermore, in the pediatric population, the overall clinical presentation is incomplete since it is still developing [30]. 

The Phenomizer tool facilitated differential diagnosis by linking candidate diseases that best corresponded to a given set of clinical features. Phenomizer produced a significant *p*-value thanks to the availability of detailed clinical data in 21 of the 97 diagnosed cases (21%), and in 2 patients with an inconclusive diagnosis, P99 and P101, in whom we identified a heterozygous pathogenic/likely pathogenic variant in a recessive gene closely related to the patient’s clinical phenotype, but no second variant was detected. Such outcomes can be difficult to achieve, given the nonspecific clinical features of some INMDs and the fact that Phenomizer requires laboratory test results and detailed clinical data for the patient in question. The Reactome pathway analysis allowed an understanding of the relationship between the INMD genes and the affected biological systems. Most of the genes analyzed in this study are clearly expressed in (a) general muscle contraction, including striated muscle (such as *ACTA1*, *DMD*, *RYR1*, *TTN*); (b) ECM physiology, including organization, degradation, ECM–proteoglycans and non-integrin membrane–ECM interactions (most of the genes included encodes collagen components, such as *COL1A2*, *COL6A1*, *COL6A2*, *COL6A3*, *COL6A6*, *COL12A1*, *COL13A1*); and (c) with nervous system development, including channelopathies (calcium channels: *CACNA1S;* sodium channels: *SCN4A*), muscular dystrophies (*LAMA2*), and myopathies (*TAZ*). Furthermore, several INMD genes were found to affect the presynaptic nicotinic acetylcholine receptors (*CHRNA1* and *CHRNE*) and metabolism, specifically glycogen metabolism (*GYG1*, *PYGM*, *PHKB*, *PHKA1*).

Large gene panels offer certain advantages over exome sequencing, providing an optimal balance between diagnostic success, diagnostic turnaround time, and cost-effectiveness. In fact, targeted gene-panel sequencing has a diagnostic rate that is 4–15% lower than that of CES or WES [22,31]. While one disadvantage of gene panels is that they do not enable the identification of new disease-associated genes, panels can be updated periodically to address this shortcoming. Moreover, a comprehensive gene panel can include high-contribution genes, thereby increasing the diagnostic yield even in cases of diseases with uncertain etiologies or variable presentations [32,33]. In our study, the most frequent mutated genes were *TTN*, *ANO5*, and *RYR1*, which together accounted for almost 30% of the diagnosed cases. The *TTN* and *RYR1* sequences commonly carry variations, which seem to be related to their large size [30]. The 364-exon *TTN* gene encodes one of the largest known proteins, titin, which is expressed in cardiac and skeletal muscle and spans half of the sarcomere from the Z line to the M line [34]. Because of its enormous size, missense variants in the *TTN* gene are relatively common in the general population [35,36]. Truncating *TTN* variants are the most common causes of dilated cardiomyopathy, occurring in 10–20% of cases [37,38]. However, the role of missense variants is less clear, but a previous study suggested their implication as a modifier of the phenotype [39]. Patients with *TTN* variants showed wide-ranging clinical presentations, from congenital-onset arthrogryposis multiplex congenita to late-onset axial muscle weakness with minicore myopathy (Appendix A). Assessment and classification of *TTN* missense variants and non-frameshifting insertions/deletions is difficult due to their high frequency [36]. Indeed, in our cohort, only one missense variant was identified: the remaining variants were indels and splicing variants, with a predominantly recessive inheritance pattern (except for P130). However, given the common presence of the *TTN* variants in the general population, large datasets are necessary to help establish an association between the variants and the clinical outcomes.

The 106-exon *RYR1* gene encodes a ryanodine receptor that is found in skeletal muscle and participates in calcium release in the sarcoplasmic reticulum (required for skeletal muscle contraction) and forms a bridging structure connecting the sarcoplasmic reticulum and transverse tubule [40]. Pathogenic *RYR1* variants can modify the structure and function of RyR1 channels, leading to mild to severe symptoms ranging from global motor delay and proximal muscle weakness to rhabdomyolysis and myopathic facies. In our cohort, we confirmed a recessive or dominant inheritance pattern in eight patients, of whom two carried de novo variants. P44 inherited both variants from their mother, who also presented myopathy and elevated serum creatine phosphokinase, while P45 inherited a missense pathogenic variant from their unaffected father, although parental mosaicism has recently been described in RYR1-related myopathies [41]. *ANO5* encodes a member of the anoctamin family of transmembrane proteins, which is likely a calcium-activated chloride channel. The 913-amino acid protein includes 8 transmembrane domains: both the N and C termini are cytoplasmic, and the extracellular regions contain 6 putative N-glycosylation sites [42]. In seven patients, we identified homozygous pathogenic variants in *ANO5*, including indel, splicing, and missense mutations. De novo heterozygous variants were detected in two patients. Interestingly, the 2 children of P91, both of whom inherited the maternal de novo variant c.1733T>C, were clinically asymptomatic but had increased serum creatine kinase levels. Although *ANO5*-associated muscular dystrophies usually follow a recessive inheritance pattern, carriers can present milder phenotypes, with clinical signs limited to hyperCKemia [43].

Advances in targeted NGS sequencing have also increased the read depth, facilitating the detection of CNVs [44,45]. In our cohort, 6% of the diagnosed patients harbored a CNV. Duchenne muscular dystrophy (MIM #310200) is commonly caused by CNVs in the *DMD* gene. We found that 2 male patients diagnosed with DMD carried a hemizygous deletion spanning exons 21–33 (P30) and a duplication of exons 24–29 (P29). In 2 other patients, we identified a homozygous (P87) and compound heterozygous (P88) deletion in exon 4 of *SIGMAR1*. Several publications have associated truncations/deletions in Sigmar1 with the development of distal hereditary motor neuropathies (MIM #605726) [46,47]. Unequal crossover due to misalignment during meiosis in the ch17p11.2 region leads to CNVs comprising the *PMP22* gene that are associated with either Charcot–Marie–Tooth (CMT1A) disease (*PMP22* duplication) or hereditary neuropathy with liability to pressure palsy (HNLP) diseases (*PMP22* deletion) [48]. Both of our patients with *PMP22* variants, one with a whole gene duplication of *PMP22* (P72) and another with duplication of exons 4 and 5 (P73), were diagnosed with CMT (MIM #118220).

In our cohort, we identified recurrent variants in 16% (16/97) of the diagnosed patients, underscoring their pathogenic effect in Spanish and Portuguese populations (recurrent variants highlighted in bold in Appendix A). The *ANO5* c.191dup variant was one of the most common in our cohort and was previously described as a founder mutation within the northern European population [49,50]. This sequence change is predicted to result in protein truncation or nonsense-mediated decay in a gene in which loss-of-function is a known mechanism of disease, and is documented as a common cause of limb-girdle muscular dystrophy. The homozygous variant c.38661_38665del in *TTN* was identified in 2 patients. In P57, a patient with arthrogryposis multiplex congenita and myopathy without cardiac involvement, the deletion was a result of maternal isodisomy for chromosome 2. This was confirmed through a study of 100 homozygous SNPs (from rs12713756 to rs12622093) distributed along chromosome 2 for the patient–parent trio. In this analysis, we detected 62 SNPs identical to those of maternal chromosome 2, potentially indicating that the deletion was a result of monosomy rescue. The most commonly detected variant in our cohort was c.2860C>T in *SH3TC2*, which was identified in 4 patients. This change creates a premature translational stop signal (*p*. Arg954*) and has been described in several European populations [51,52] and reported as a founder mutation in the French Canadian population [53].

The diagnostic success of NGS in INMD is linked not only to the panel design but also to the stringency of the cohort inclusion criteria [54]. Although we did not establish criteria stringency in this study, apart from suspicion of a neuromuscular disorder, we found that 68% of the diagnosed patients had a molecular diagnosis of muscular dystrophy/myopathy and 23% had a molecular diagnosis of peripheral nerve disease. Targeted NGS sequencing enables rapid analysis of INMD-related genes in pediatric and adult patients, providing reliable genetic results rapidly and cost effectively while reducing the likelihood of incidental findings. The identification of INMDs in adulthood can be difficult, as by this stage, the disease has usually progressed, and therefore the symptoms at onset cannot be identified. In children, a positive molecular diagnosis can avoid the need for invasive diagnostic processes, such as muscle biopsy. 

One of the biggest challenges of targeted gene analysis is the design of a panel that maximizes the diagnostic yield. Our findings suggest that a smaller disease-specific panel would have been less effective, since our diagnostic yield increased exponentially with the successive addition of genes to the panel design. Other challenges associated with NGS analysis include the interpretation of VUS, the absence of family history, and overlapping NMD phenotypes. Indeed, ACMG guidelines can guide clinical geneticists to ascertain the pathogenicity of a specific variant; however, some of these criteria should be assessed carefully. For instance, most of the individuals included in the Genome Aggregation Database (GnomAD) are adults with a mean age of 54 years and attempts have been made to exclude individuals with severe pediatric diseases and their first-degree relatives [55]. These variables complicate interpretation of the molecular findings, and often necessitate further studies. Furthermore, NGS approaches can fail to detect variants in coding (due to decreased coverage in GC-rich and highly homologous regions) and non-coding regions, including changes that affect RNA expression and splicing. The detection of these changes usually requires additional functional studies at the transcriptional level. In fact, in seven of our patients, we identified a likely pathogenic variant in a recessive gene closely related to the patient’s clinical phenotype but did not detect a second variant (Appendix A). The utility of RNA-seq for solving undiagnosed cases by detecting disease-causing variants associated with neuromuscular disorders remains to be explored [56].

## 4. Materials and Methods

### 4.1. Study Design

This prospective multicenter study was conducted over a 3-year period in the Metabolic Unit of the University Clinical Hospital of Santiago de Compostela (Spain), in coordination with 10 other centers in Spain and Portugal. Informed consent was obtained prior to genetic analysis from the patient or the patient’s legal guardians. This study was approved by the Clinical Research Ethics Committee of Galicia (Reference code 2015/410). 

### 4.2. Study Population

Patients aged 0–80 years with a suspected INMD were included in this study. Patients were clustered into those with symptoms compatible with muscular/myopathies, peripheral nerve diseases, neuromuscular junction diseases/ion channel dystrophies diseases, and motor neuron diseases. The relevant clinical information, including clinical symptoms and complementary data, were collected using Human Phenotype Ontology (HPO) terms (https://hpo.jax.org/, accessed on 28 February 2022) [57]. All HPO terms for each patient with a positive diagnosis are described in Appendix A.

The targeted analysis was the first genetic test performed for all the patients included in our cohort. For confirmation of disease segregation, family studies were performed whenever possible through Sanger sequencing to determine the inheritance pattern. Analysis of the HPO terms using the Phenomizer tool allowed us to correlate the phenotype with the obtained molecular diagnosis (https://compbio.charite.de/phenomizer/, accessed on 27 May 2021) [58]. 

### 4.3. Gene Panel Design

We designed a multi-gene panel consisting of a group of 324 genes previously described in the literature and associated with INMD, including muscular dystrophies, muscular myopathies, peripheral nerve disease, motor neuron disease, ion channel diseases, and neuromuscular junction diseases. This multi-gene panel was updated throughout the 3-year study period by reviewing the most recent scientific publications to include new genes potentially associated with INMD. Several versions of the panel were designed and are available upon request (final version of the INMD panel is available in Appendix A).

### 4.4. Genetic Analysis

Gene-targeted analysis was performed by in-solution hybridization technology (Sure Select XT; Agilent Technologies, Santa Clara, California, USA), followed by sequencing using the Miseq platform (Illumina, San Diego, California, USA). A custom Sure Select probe library was designed to capture the exons and exon-intron boundaries of the targeted genes [59]. Base calling was performed with Real Time Analysis (RTA) software v.1.8.70 (Illumina, San Diego, California, USA), and the FastQC v0.10.1 program was used for data quality control. Reads were aligned to the reference genome GRCh37 with BWA v0.7.9a [60]. Variant detection was performed with VarScan v.2.3.6 [61] and SAMtools v0.1.19 [62] for indels and SNP, respectively, and Annovar for variant annotation [63]. For CNV detection, we used PattRec, an optimized CNV detection tool for targeted-NGS data [64]. 

To ensure reliable clinical interpretation of the variants detected, we applied prioritization criteria to predict pathogenicity according to the ACMG guidelines [55,65]. The variants were analyzed *in silico* to determine the evolutionary conservation, functional consequences, and minor allele frequency (MAF) within the population. We used the algorithms Genomic Evolutionary Rate Profiling (GERP) [66], PhyloP [67], and phastCons [68] to determine if the variants identified were highly conserved through evolution. Pathogenicity was predicted as disease causing by MutationTaster [69] and damaging by the FATHMM [70] and DANN scores [71]. The Genome Aggregation Database (GnomAD) v2.1.1 data set was applied to infer specific variant allele frequencies in the population [72]. Detailed descriptions and pathogenicity classification for each variant are available in Appendix A.

### 4.5. Statistical Analysis

The Phenomizer algorithm was used to compare the clinical features of each patient against those of a set of annotated diseases, ranked according to their *p*-values. *p*-values were adjusted for multiple testing using the Benjamin–Hochberg method [73]. A *p*-value < 0.05 was considered statistically significant.

## 5. Conclusions

In conclusion, our results confirm that while clinical characteristics and histopathologic features of INMDs can suggest a specific INMD, achieving a definitive diagnosis requires the identification of pathogenic variants in candidate disease-causing genes. Using gene-panel sequencing, we were able to identify 6 recurrent and 28 de novo variants, allowing us to rapidly establish a definitive genetic diagnosis in 97 patients. As a first-tier strategy, we recommend comprehensive gene-panel analysis of all neuromuscular disease-related genes, including those commonly implicated in these diseases (e.g., *TTN* and *RYR1*). This approach facilitates the diagnosis of previously undiagnosed patients while providing a high diagnostic rate.

## Figures and Tables

**Figure 1 jcm-11-02750-f001:**
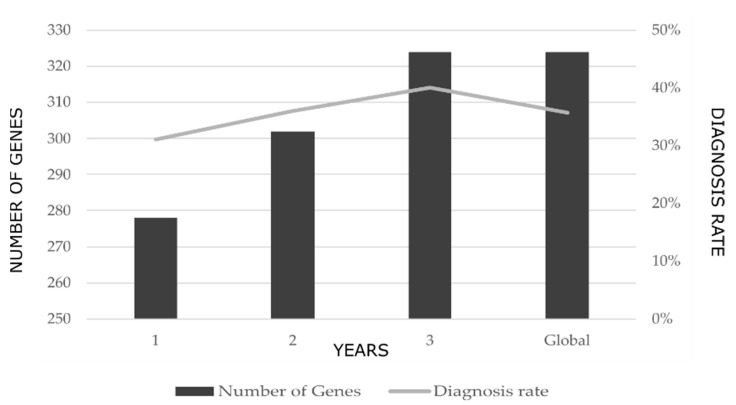
Change in the rate of diagnosis with the successive addition of genes to the multi-gene panel over the 3-year study period. The global value represents the mean rate of diagnosis over the entire study period.

**Figure 2 jcm-11-02750-f002:**
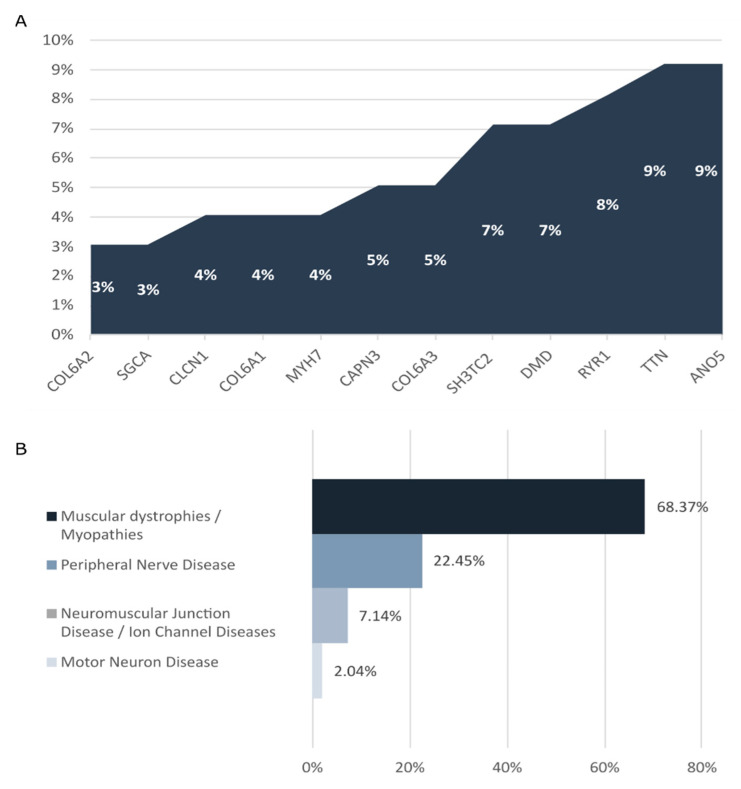
(**A**) Genes detected with high frequency in diagnosed patients analyzed with the INMD panel. Graph depicts the proportion of clinical cases for which a pathogenic/likely pathogenic variant was identified in the causative gene. (**B**) INMDs identified in our cohort.

**Figure 3 jcm-11-02750-f003:**
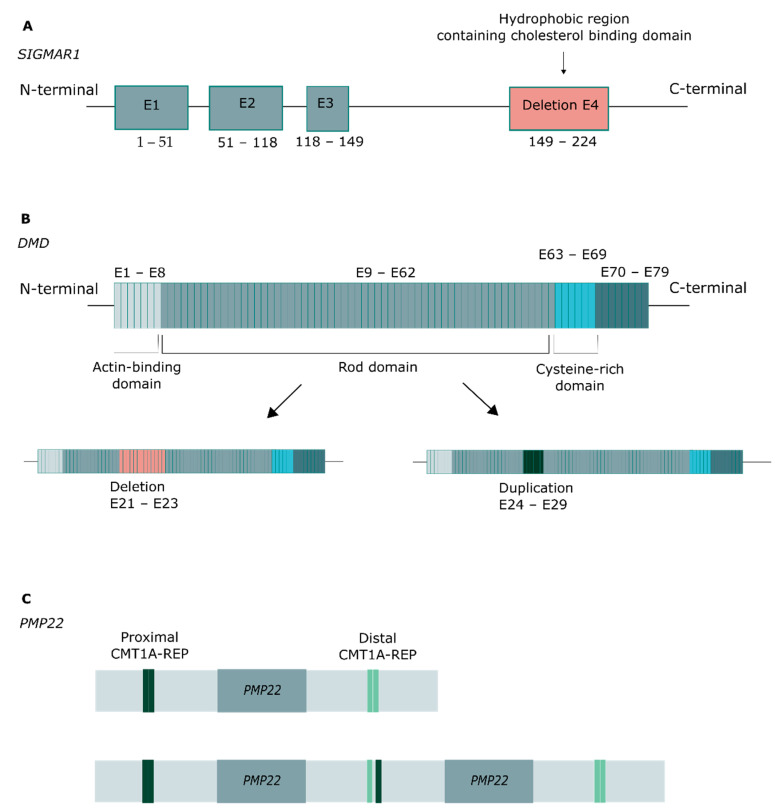
CNVs identified in the study population. (**A**) Exon 4 deletion in SIMGAR1, affecting the hydrophobic region. (**B**) Deletion and duplication in DMD, affecting the rod domain. (**C**) Duplication of the entire PMP22 gene, associated with Charcot–Marie–Tooth disease.

**Figure 4 jcm-11-02750-f004:**
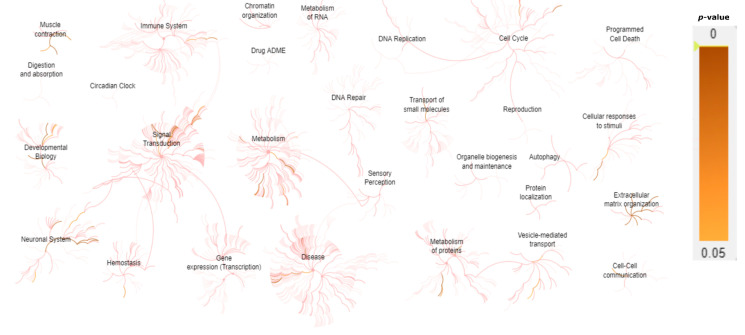
Representation of the predicted biological pathways associated with INMD genes included in the gene-panel design. Gene expression values (*p*-value < 0.05) are represented using Reactome pathway enrichment analysis. The majority of the genes included in the INMD gene panel are predicted to be highly expressed in muscle contraction, neuronal system, signal transduction, metabolism, cell cycle, ECM, vesicle-mediated transport, and cellular responses to stimuli.

**Table 1 jcm-11-02750-t001:** Representation of the predicted biological systems affected due to pathogenic variants in the associated genes, leading to the development of INMD, using the Reactome pathway database.

Biological System	Associated Genes
Muscle contraction	*ACTA1*, *ATP2A1*, *CASQ1*, *CAV3*, *DES*, *DMD*, *DYSF*, *KCNJ12*, *MYBPC1*, *MYH3*, *MYH8*, *NEB*, *ORAI1*, *RYR1*, *SCN4A*, *SCN10A*, *STIM1*, *TAZ*, *TCAP*, *TDP1*, *TNNT1*, *TNNT3*, *TNNI2*, *TPM2*, *TPM3*, *TTN*
Glycogen metabolism	*AGL*, *GAA*, *GBE1*, *GYG1*, *GYS1*, *PYGM*, *PHKB*, *PHKA1*
Extracellular matrix organization/ degradation	*AGRN*, *CAPN3*, COL1A2, COL6A1, COL6A2, COL6A3, COL6A6, COL12A1, *COL13A1*, *DAG1*, *DMD*, *DST*, *FBN2*, *FBLN5*, *ITGA7*, *LAMA2*, *LAMB2*, *LRP4*, *MME*, *MUSK*, *PLEC*, *TNXB*, *TTR*
O-linked glycosylation	*B3GNT2*, *B3GALNT2*, *DAG1*, *LARGE*, *POMK*, *POMGNT1*, *POMGNT2*, *POMT1*, *POMT2*
NCAM signaling for neurite out-growth/ interactions	*AGRN*, *HRAS*, *CACNA1S*, *COL6A1*, *COL6A2*, *COL6A3*, *COL6A6*, *SPTBN4*
EGR2 and SOX10-mediated initiation of Schwann cell myelination	*ADGRG6*, *DAG1*, *DRP2*, *EGR2*, *GJB1*, *PRX*, *LAMA2*, *LAMB2*, *MPZ*, *PMP22*, *TAZ*
tRNA Aminoacylation	*AARS*, *DARS2*, *GARS*, *HARS*, *KARS*, *MARS*, *WARS*, *YARS*, *YARS2*
Nervous system development	*ADGRG6*, *AGRN*, *CACNA1S*, *CNTN1*, *CNTNAP1*, *COL6A1*, *COL6A2*, *COL6A3*, *COL6A6*, *DAG1*, *DNM2*, *DRP2*, *EGR2*, *GJB1*, *HRAS*, *LAMA2*, *LAMB2*, *MPZ*, *MYH14*, *PIP5K1C*, *PMP22*, *PRX*, *SCN4A*, *SCN10A*, *SPTBN4*, *TAZ*
Presynaptic nicotinic acetylcholine receptors	*CHRNA1*, *CHRND*, *CHRNE*, *CHRNG*

## Data Availability

The data presented in this study are available in Appendix A.

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
