# Peer review of "Rapid Molecular Diagnosis of Genetically Inherited Neuromuscular Disorders Using Next-Generation Sequencing Technologies"

_jcm, 2022, doi:10.3390/jcm11102750_

Round 1
Reviewer 1 Report
The authors carried out a resequencing study including 268 patients with inherited neuromuscular disorders (INMDs) and here describe the main findings of their study. Although the novelty is limited (similar studies have been already carried out 6-7 years ago and with similar results), the article is overall interesting and well written.
However, I have some main concerns (mainly related to the variant interpretation) I would like to share with the authors.
1) The cohort should be better outlined to fully appreciate the results.
a) Is the described targeted analysis the first performed test for all these patients? This of course affects the detection rate.
b) The authors write that: ´Patients were clustered into those with symptoms compatible with muscular dystrophies (limb-girdle, congenital Ullrich, Duchenne, Emery-Dreifuss), congenital myopathies, distal myopathies, metabolic myopathies, neuromuscular junction diseases, spinal amyotrophy, and hereditary neuropathies´.
Supplementary Tables list the HPO terms for the enrolled patients. However, the Table does not fully reflect the clusters cited in the main text.
Moreover, in line with previous studies, it could be useful to show the detection rate for each of the aforementioned categories (including the subcategories, if it is doable) and even discuss the possible different detection rates.
I find very interesting the relatively high number of de novo variants identified. It could be nice to show that, as expected, de novo variants mainly occur in patients with a congenital or early onset (and the lack of parental DNA could have hampered the identification of de novo variants in previous studies).
c) The authors should clarify if patients were always analyzed as singletons and, thereby, the following family studies are always Sanger-based. This is also useful to evaluate the turn-around time (´In cases in which a positive molecular diagnosis was achieved, the diagnostic turnaround time was 4–6 weeks´).
d) Has a CNV analysis been performed for the entire cohort? Material and Methods section does not mention the CNV tools used for the analysis. Moreover, considering the technical issues related to CNV-tools, it would be interesting to describe how the results were filtered.
2) Results:
a) For all the variants, also the protein change should be reported so that readers can distinguish missense variants from nonsense variants in the Supplementary Tables. Similarly, previously described causative variants could be tagged.
b) The authors write: ´To ensure a reliable clinical interpretation of the variants detected, we applied prioritization criteria to predict pathogenicity according to ACMG guidelines´. This is a very important point and, thereby, for all the variants, the ACMG classification should be reported in Supp Tables.
c) For each gene, the reference transcript should be indicated.
Although the lack of a full annotation (i.e., protein annotation, gnomAD frequency and ACMG classification) does not allow me to carefully recheck the Supplementary Tables, I still noticed the following unclear points:
- a) ANO5 findings:
For ANO5, authors indicate, as a related phenotype, a muscular dystrophy (# 611307), defined as AD/AR (Supp Table 1).
However, ANO5-related muscular dystrophy is fully recessive (and the OMIM code # 611307 refers to a recessive condition). Although several studies (Pentila et al 2012; Savarese et al, 2015 and more recent ones) suggest that variants in heterozygosity may result in hyperckemia and a mild phenotype, this has never been fully demonstrated. Thereby, the identification of a single variant (even if it is a de novo) does not really result in a confirmed diagnosis (it could be considered an interesting finding but for sure not a definitive diagnosis – in particular using the ACMG criteria - also because of the lack of functional data on ANO5 variants).
b) Similar issue with titin (TTN). Few TTN-related muscle diseases are dominant. And, so far, truncating variants have been mostly reported in recessive conditions (and, as expected, most of your patients have bi-allelic truncating variants with healthy carrier parents).
It is thereby difficult to explain how the variant c.91615_91616dup in pat.68 causes the observed phenotype in heterozygosity.
Similarly, the discussion on the titin-related diseases is very general and lacks some crucial points. The currently indicated references (30 and 31) do not refer to titin-related muscle diseases. The recent studies focusing on the interpretation of titin variants (including missense variants) in skeletal muscle titinopathies and a possible genotype-phenotype correlation should be properly discussed.
c) Similarly, in pat 20, the COL6 variant is inherited from the father. Does this mean that the father is also affected?
I listed above only few examples of somehow ´conflicting interpretations´. However, it is well known that, despite the efforts, variant interpretation largely differs among different labs and this heavily affects the detection rate. This could be further discuss. Moreover, the discussion should also exhaustively cover the previous literature on the diagnostic rate in INMDs.
Among the others, (some of) the following studies could be discussed to further delineate the elements affecting the diagnostic rate and hampering a higher success rate:
- Ankala A, da Silva C, Gualandi F, et al. A comprehensive genomic approach for neuromuscular diseases gives a high diagnostic yield. Ann Neurol 2015; 77:206–214.
-Chae JH, Vasta V, Cho A, et al. Utility of next generation sequencing in genetic diagnosis of early onset neuromuscular disorders. J Med Genet 2015; 52:208–216.
- Evila A, Arumilli M, Udd B, Hackman P. Targeted next-generation sequencing assay for detection of mutations in primary myopathies. Neuromuscul Disord 2016; 26:7–15.
- Kuhn M, Glaser D, Joshi PR, et al. Utility of a next-generation sequencing-based gene panel investigation in German patients with genetically unclassified limb-girdle muscular dystrophy. J Neurol 2016; 263:743–750.
- Savarese M, Di Fruscio G, Torella A, et al. The genetic basis of undiagnosed muscular dystrophies and myopathies: results from 504 patients. Neurology 2016; 87:71–76.
- Sevy A, Cerino M, Gorokhova S, et al. Improving molecular diagnosis of distal myopathies by targeted next-generation sequencing. J Neurol Neurosurg Psychiatry 2016; 87:340–342.
It could be also interesting to reflect on how the panel design and the variant interpretation has evolved in the last few years.
Minor comments:
Supp Table 4: it could be useful to specify the genes included in each specific version
Digenic inheritance: I would define these very interesting cases as possible dual diagnoses or Multilocus Genomic Variation. This topic (that is relatively new and poorly covered in the previous studies) could be further discussed (Posey et al. 2017; Jones et al. 2018)
Author Response
Comments and Suggestions for Authors: The authors carried out a resequencing study including 268 patients with inherited neuromuscular disorders (INMDs) and here describe the main findings of their study. Although the novelty is limited (similar studies have been already carried out 6-7 years ago and with similar results), the article is overall interesting and well written. However, I have some main concerns (mainly related to the variant interpretation) I would like to share with the authors.
ANSWER: We would like to thank the reviewer for a careful and thorough reading of our manuscript and for the constructive suggestions, which improved the quality of the manuscript.
1) The cohort should be better outlined to fully appreciate the results.
- a) Is the described targeted analysis the first performed test for all these patients? This of course affects the detection rate.
ANSWER: This is correct, the targeted analysis was the first genetic test performed for all the patients included in our cohort. This information was added in the Material and Methods section (lines 339-340): The targeted analysis was the first genetic test performed for all the patients included in our cohort.
- b) The authors write that: ´Patients were clustered into those with symptoms compatible with muscular dystrophies (limb-girdle, congenital Ullrich, Duchenne, Emery-Dreifuss), congenital myopathies, distal myopathies, metabolic myopathies, neuromuscular junction diseases, spinal amyotrophy, and hereditary neuropathies´. Supplementary Tables list the HPO terms for the enrolled patients. However, the Table does not fully reflect the clusters cited in the main text. Moreover, in line with previous studies, it could be useful to show the detection rate for each of the aforementioned categories (including the subcategories, if it is doable) and even discuss the possible different detection rates.
ANSWER: You are right, this information was not clear in the manuscript. This was corrected (lines 332-335): Patients were clustered into those with symptoms compatible with muscular dystrophies/ myopathies, peripheral nerve diseases, neuromuscular junction diseases/ ion channel diseases, and motor neuron diseases.
Moreover, according to the reviewer, the detection rate for each category was added to the manuscript in the Results section (lines 129-132): The diagnostic rate achieved in each INMD group was as follows: 46.53% in muscular dystrophies/myopathies, 30.99% in peripheral nerve diseases, 18.91% in neuromuscular junction diseases/ ion channel diseases, and 12.5% in motor neuron diseases.
- c) I find very interesting the relatively high number of de novo variants identified. It could be nice to show that, as expected, de novo variants mainly occur in patients with a congenital or early onset (and the lack of parental DNA could have hampered the identification of de novo variants in previous studies).
ANSWER: We agree with the reviewer and this was clarified in the discussion (lines 194-196): This high rate might be associated with an early onset of the disorder (18 patients with a de novo variant was under 20 years old) and the possibility to perform family studies.
- d) The authors should clarify if patients were always analyzed as singletons and, thereby, the following family studies are always Sanger-based. This is also useful to evaluate the turn-around time (´In cases in which a positive molecular diagnosis was achieved, the diagnostic turnaround time was 4–6 weeks´).
ANSWER: On average we had a complete genetic sequencing analysis in 4 weeks, and 2 more weeks were needed to complete family studies through Sanger analysis. In the Material and Methods section, we have added the following sentence (lines 340-341): For confirmation of disease segregation, family studies were performed whenever possible through Sanger sequencing to determine the inheritance pattern.
- e) Has a CNV analysis been performed for the entire cohort? Material and Methods section does not mention the CNV tools used for the analysis. Moreover, considering the technical issues related to CNV-tools, it would be interesting to describe how the results were filtered.
ANSWER: CNV analysis was performed for the entire cohort. Thank you for the comment, this information was missing in the Material and Methods section and was added for better understanding (lines 364-365): For CNV detection we have used PattRec, an optimized CNV detection tool for targeted-NGS data [55].
2) Results:
- a) For all the variants, also the protein change should be reported so that readers can distinguish missense variants from nonsense variants in the Supplementary Tables. Similarly, previously described causative variants could be tagged.
ANSWER: According to the reviewer, detailed information on the variants identified in this work and the pathogenic classification according to ACMG guidelines was added in Supplementary table S4.
- b) The authors write: ´To ensure a reliable clinical interpretation of the variants detected, we applied prioritization criteria to predict pathogenicity according to ACMG guidelines´. This is a very important point and, thereby, for all the variants, the ACMG classification should be reported in Supp Tables.
ANSWER: This information was added in Supplementary table S4.
- c) For each gene, the reference transcript should be indicated.
ANSWER: Information was added in Supplementary table S4.
- d) Although the lack of a full annotation (i.e., protein annotation, gnomAD frequency and ACMG classification) does not allow me to carefully recheck the Supplementary Tables, I still noticed the following unclear points:
- ANO5 findings:
For ANO5, authors indicate, as a related phenotype, a muscular dystrophy (# 611307), defined as AD/AR (Supp Table 1). However, ANO5-related muscular dystrophy is fully recessive (and the OMIM code # 611307 refers to a recessive condition). Although several studies (Pentila et al 2012; Savarese et al, 2015 and more recent ones) suggest that variants in heterozygosity may result in hyperckemia and a mild phenotype, this has never been fully demonstrated. Thereby, the identification of a single variant (even if it is a de novo) does not really result in a confirmed diagnosis (it could be considered an interesting finding but for sure not a definitive diagnosis – in particular using the ACMG criteria - also because of the lack of functional data on ANO5 variants).
ANSWER: Thanks for the correction, indeed ANO5 associated with muscular dystrophy has a recessive inheritance pattern. This mistake led us to review the supplementary tables. ANO5 inheritance pattern defined as AD/AR, was corrected to AR in Supplementary Tables S1 and S2.
ANSWER: Patients with heterozygous variants in ANO5 and TTN were initially included in the diagnosed group since we had a diagnostic confirmation from the clinician. However, we do agree that in these cases molecular studies are needed to confirm the genotype-phenotype correlation. For this reason, these cases were deleted from Supp Table S1, and added in Supp Table S2 and S3, a group of patients with inconclusive diagnoses (P91 and P130). We do analyze the data again and for the same reason, other patients were considered inconclusive diagnoses: the diagnostic rate was updated to 33% (88/268 patients).
- Similar issue with titin (TTN). Few TTN-related muscle diseases are dominant. And, so far, truncating variants have been mostly reported in recessive conditions (and, as expected, most of your patients have bi-allelic truncating variants with healthy carrier parents). It is thereby difficult to explain how the variant c.91615_91616dup in pat.68 causes the observed phenotype in heterozygosity. Similarly, the discussion on the titin-related diseases is very general and lacks some crucial points. The currently indicated references (30 and 31) do not refer to titin-related muscle diseases. The recent studies focusing on the interpretation of titin variants (including missense variants) in skeletal muscle titinopathies and a possible genotype-phenotype correlation should be properly discussed.
ANSWER: Thank you for the suggestion, we have extended the discussion regarding TTN variants (lines228-232): Because of its enormous size, missense variants in the TTN gene, are relatively common in the general population [34,35]. Truncating TTN variants are the most common causes of dilated cardiomyopathy, occurring in 10–20% of cases [36,37]. However, the role of missense variants is less clear but a previous study suggested their implication as a modifier of the phenotype [38].
(lines 223-241): However, given the common presence of the TTN variants in the general population, it is necessary large datasets to help setting an association of the variants with the clinical outcomes.
- Similarly, in pat 20, the COL6 variant is inherited from the father. Does this mean that the father is also affected?
ANSWER: In patient 20, the variant was inherited from the affected father. For better understanding, the terms “affected” and “unaffected” were added in the Supplementary Material Table S1, in those cases where the variant was inherited from one of the parents.
- e) I listed above only few examples of somehow ´conflicting interpretations´. However, it is well known that, despite the efforts, variant interpretation largely differs among different labs and this heavily affects the detection rate. This could be further discuss. Moreover, the discussion should also exhaustively cover the previous literature on the diagnostic rate in INMDs. Among the others, (some of) the following studies could be discussed to further delineate the elements affecting the diagnostic rate and hampering a higher success rate:
- Ankala A, da Silva C, Gualandi F, et al. A comprehensive genomic approach for neuromuscular diseases gives a high diagnostic yield. Ann Neurol 2015; 77:206–214.
-Chae JH, Vasta V, Cho A, et al. Utility of next generation sequencing in genetic diagnosis of early onset neuromuscular disorders. J Med Genet 2015; 52:208–216.
- Evila A, Arumilli M, Udd B, Hackman P. Targeted next-generation sequencing assay for detection of mutations in primary myopathies. Neuromuscul Disord 2016; 26:7–15.
- Kuhn M, Glaser D, Joshi PR, et al. Utility of a next-generation sequencing-based gene panel investigation in German patients with genetically unclassified limb-girdle muscular dystrophy. J Neurol 2016; 263:743–750.
- Savarese M, Di Fruscio G, Torella A, et al. The genetic basis of undiagnosed muscular dystrophies and myopathies: results from 504 patients. Neurology 2016; 87:71–76.
- Sevy A, Cerino M, Gorokhova S, et al. Improving molecular diagnosis of distal myopathies by targeted next-generation sequencing. J Neurol Neurosurg Psychiatry 2016; 87:340–342.
ANSWER: Thank you very much for your comments, we have included several suggested references in the Discussion section regarding the diagnostic rates in gene-targeted NGS analysis (line 180). We have also expanded the discussion reflecting the results of diagnostic rates of other studies due to the conflicting interpretations between different groups (lines 181-193): Despite ACMG guidelines to better classify pathogenic variants, there are still conflicting interpretations among different genetic laboratories which in turn affects diagnostic rate. In a recent study, Winder et al. aimed to demonstrate the clinical utility of a targeted-gene panel including 266 genes by analyzing 25,356 unrelated patients with a suspicion of INMD. A definitive diagnosis was achieved in 5,055 (20%) of the patients, and the CNVs detection accounted for 39% of the variants identified [27]. Savarese et al. used a NGS plat-form named MotorPlex, including 93 genes associated with nonsyndromic myopathies which usually cannot be clinically diagnosed. Five hundred and four patients and eighty-four family members from the Italian Network of Congenital Myopathies and the Italian Network of Limb-Girdle Muscular Dystrophy were studied and as a result, 218 (43.3%) cases obtained a positive diagnosis, and 160 patients had undetermined diagnosis since interesting candidate variants were identified, but unproven [28].
- f) It could be also interesting to reflect on how the panel design and the variant interpretation has evolved in the last few years.
ANSWER: A retrospective analysis of variant identification/interpretation was not applied in this study. Although several published studies have impressive results in the diagnostic rates when a reanalysis of previously identified variants, mainly VUS variants, is performed; in our cohort, the increased diagnostic rate is along with the increase of the number of genes included in the design panel. Indeed, internally we have some ongoing studies in the reinterpretation of VUS variants that can change the previously reported diagnostic rates.
Minor comments:
-Supp Table 4: it could be useful to specify the genes included in each specific version
ANSWER: In the Material and Methods section (lines 351-353): Several versions of the panel were designed and are available upon request (the final version of the INMD panel in supplementary Table S4). However, if the reviewer considers it highly important to add the different INMD panels versions in the Supplementary Material, we can add this information.
-Digenic inheritance: I would define these very interesting cases as possible dual diagnoses or Multilocus Genomic Variation. This topic (that is relatively new and poorly covered in the previous studies) could be further discussed (Posey et al. 2017; Jones et al. 2018)
ANSWER: We agree with the suggestion. We have updated this information in the manuscript as follows:
Results (lines 107-108): Two possible cases of dual diagnosis were suspected.
Discussion (line 168): Dual diagnosis was suspected in 6% (2/35) of cases.
Reviewer 2 Report
Thank you for the paper using NGS to quickly detect INMD. You clearly explained you tested INMD related genes in your population suspected for INMD. You nicely discuss several associated genes. To me its not solid enough for this journal.
What my main concern is which should be addressed in my opinion is the following: All data are tested in a potentially affected population. This implies we need to assume the normal population (non INMD “suspects”) should give highly contrasting findings. To truly show these marker genes are of value I now miss this “normal” population as comparison. This makes it hard to say these genes are involved except for the proven ones like the premature stopcodon in SH3TC2 many are currently associations. This also maybe highly population dependent.
Secondly what you normally see in a paper like this is a full scheme of potential genes involved and their pathways (including their interconnectivity). This shows which genes may be involved and should be tested as well (although they may not yet be known). By first building this network using the genes you already know to be involved and then do your analysis, the paper starts become way more interesting to read.
Author Response
Comments and Suggestions for Authors: Thank you for the paper using NGS to quickly detect INMD. You clearly explained you tested INMD related genes in your population suspected for INMD. You nicely discuss several associated genes. To me its not solid enough for this journal.
ANSWER: We would first like to thank the reviewer for a careful and thorough reading of our manuscript and for the constructive suggestions.
- What my main concern is which should be addressed in my opinion is the following: All data are tested in a potentially affected population. This implies we need to assume the normal population (non INMD “suspects”) should give highly contrasting findings. To truly show these marker genes are of value I now miss this “normal” population as comparison. This makes it hard to say these genes are involved except for the proven ones like the premature stopcodon in SH3TC2 many are currently associations. This also maybe highly population dependent.
ANSWER: Thank you very much for the suggestion but the main purpose of this study was to evaluate the diagnostic yield and utility of an INMD gene-panel to understand the mutational architecture of those genes associated with these disorders. With this, strictly speaking of diagnostic performance, we understand that samples should be from affected patients in whom, due to their clinical symptoms and suspicion of INMD, is requested targeted-NGS. Doing this analysis in healthy controls we do not believe it could be useful, since we would be analyzing controls that we know will be negative. What the panel already includes are genes associated with INMD, but this association is not being assessed, which is where we could understand that it would be useful to check controls. This is not the case, in fact one of the pathogenicity criteria of the variants according to the ACMG is their absence in healthy controls. As a suggestion of reviewer 1, we have added Supplementary Table S4 with the different ACMG criteria for each variant identified in this study. One of the criteria, PM2 refers to: "Absent from controls (or at extremely low frequency if recessive) in Exome Sequencing Project, 1000 Genomes Project, or Exome Aggregation Consortium". In fact, more than 90% of the variants identified in this study meets that specific criteria.
- Secondly what you normally see in a paper like this is a full scheme of potential genes involved and their pathways (including their interconnectivity). This shows which genes may be involved and should be tested as well (although they may not yet be known). By first building this network using the genes you already know to be involved and then do your analysis, the paper starts become way more interesting to read.
ANSWER: Although we agree with your suggestion, it is not the main goal of this work to establish functional relationships between the selected genes associated with INMD. We understand that this relationship does not have to exist at the strictly functional level of the gene. The NGS analysis of each of the patients included in the study attempts to establish a molecular diagnosis, regardless of the genes included in the analysis. On the other hand, we associate specific genes with specific groups of INMD (muscular dystrophies/ myopathies, peripheral nerve diseases, neuromuscular junction diseases/ ion channel diseases and motor neuron diseases), and this is detailed in our manuscript (Supplementary Material Table S1).
However, we made substantial changes to the article based also on reviewer 1's suggestions:
- In the results section we have added the detection rate for each INMD category
- Detailed information of the variants identified and the pathogenic classification according to ACMG guidelines was added in the Supplementary table S4.
- Improved the Discussion about the detection of de novo variants identified in our cohort
- Improved the discussion on TTN gene variants
- Improved the discussion of variant interpretation which largely differs among different labs and affects the detection rate of INMD
Round 2
Reviewer 1 Report
The authors provided a comprehensive reply to the reviewers´ comments. The revised version well described the methods and the results and the discussion is overall convincing.
Author Response
Thank you very much for helping improve the manuscript.
Reviewer 2 Report
- What my main concern is which should be addressed in my opinion is the following: All data are tested in a potentially affected population. This implies we need to assume the normal population (non INMD “suspects”) should give highly contrasting findings. To truly show these marker genes are of value I now miss this “normal” population as comparison. This makes it hard to say these genes are involved except for the proven ones like the premature stopcodon in SH3TC2 many are currently associations. This also maybe highly population dependent.
ANSWER: Thank you very much for the suggestion but the main purpose of this study was to evaluate the diagnostic yield and utility of an INMD gene-panel to understand the mutational architecture of those genes associated with these disorders. With this, strictly speaking of diagnostic performance, we understand that samples should be from affected patients in whom, due to their clinical symptoms and suspicion of INMD, is requested targeted-NGS. Doing this analysis in healthy controls we do not believe it could be useful, since we would be analyzing controls that we know will be negative.
<<That is my whole point this should be negative though now we do not know whether these patterns are present in the “normal” population. In other words, -except for the proven ones- we now may chase for patterns that do not tell us things but is just a bias of your specific population. The proven ones may be just “in linkage” with the ones you here test while they are not truly relevant.>>
What the panel already includes are genes associated with INMD, but this association is not being assessed, which is where we could understand that it would be useful to check controls. This is not the case, in fact one of the pathogenicity criteria of the variants according to the ACMG is their absence in healthy controls. As a suggestion of reviewer 1, we have added Supplementary Table S4 with the different ACMG criteria for each variant identified in this study. One of the criteria, PM2 refers to: "Absent from controls (or at extremely low frequency if recessive) in Exome Sequencing Project, 1000 Genomes Project, or Exome Aggregation Consortium". In fact, more than 90% of the variants identified in this study meets that specific criteria.
- Secondly what you normally see in a paper like this is a full scheme of potential genes involved and their pathways (including their interconnectivity). This shows which genes may be involved and should be tested as well (although they may not yet be known). By first building this network using the genes you already know to be involved and then do your analysis, the paper starts become way more interesting to read.
ANSWER: Although we agree with your suggestion, it is not the main goal of this work to establish functional relationships between the selected genes associated with INMD. We understand that this relationship does not have to exist at the strictly functional level of the gene. The NGS analysis of each of the patients included in the study attempts to establish a molecular diagnosis, regardless of the genes included in the analysis. On the other hand, we associate specific genes with specific groups of INMD (muscular dystrophies/ myopathies, peripheral nerve diseases, neuromuscular junction diseases/ ion channel diseases and motor neuron diseases), and this is detailed in our manuscript (Supplementary Material Table S1).
<<I see I still would appreciate a clear connected scheme to make logic out of the data. Otherwise, it seems just a stack of genes to be taken into account to detect INMD. If that is the case why this way of analyzing and not using a machine learning program to analyze the whole dataset against validated -proven- diagnoses?>>
However, we made substantial changes to the article based also on reviewer 1's suggestions:
- In the results section we have added the detection rate for each INMD category
- Detailed information of the variants identified and the pathogenic classification according to ACMG guidelines was added in the Supplementary table S4.
- Improved the Discussion about the detection of de novo variants identified in our cohort
- Improved the discussion on TTN gene variants
- Improved the discussion of variant interpretation which largely differs among different labs and affects the detection rate of INMD
<<Thank you this indeed improves though the above-mentioned suggestions remain in place as they don’t answer the question but answers the reason that this is not done.>>
Author Response
- <<That is my whole point this should be negative though now we do not know whether these patterns are present in the “normal” population. In other words, -except for the proven ones- we now may chase for patterns that do not tell us things but is just a bias of your specific population. The proven ones may be just “in linkage” with the ones you here test while they are not truly relevant.>>
ANSWER: We understand what you mean, but this study is for diagnosis of the affected population and the gene panel would be negative in the normal population.
- <<I see I still would appreciate a clear connected scheme to make logic out of the data. Otherwise, it seems just a stack of genes to be taken into account to detect INMD. If that is the case why this way of analyzing and not using a machine learning program to analyze the whole dataset against validated -proven- diagnoses?>>
ANSWER: We agree that this scheme might highlight some known genes associated with INMD and we have included this in the supplementary material. In the Material and Methods section, we have added the following (lines 348 – 350): functional protein association of the INMD-genes network is available in supplementary Figure S1, as well as a list of the genes for each INMD group on supplementary Table S6.
- <<Thank you this indeed improves though the above-mentioned suggestions remain in place as they don’t answer the question but answers the reason that this is not done.>>
ANSWER: Thank you very much for the understanding.
This manuscript is a resubmission of an earlier submission. The following is a list of the peer review reports and author responses from that submission.